# Ebola Virus Disease Vaccines: Development, Current Perspectives & Challenges

**DOI:** 10.3390/vaccines11020268

**Published:** 2023-01-26

**Authors:** Sumira Malik, Shristi Kishore, Sagnik Nag, Archna Dhasmana, Subham Preetam, Oishi Mitra, Darwin A. León-Figueroa, Aroop Mohanty, Vijay Kumar Chattu, Marjan Assefi, Bijaya K. Padhi, Ranjit Sah

**Affiliations:** 1Amity Institute of Biotechnology, Amity University Jharkhand, Ranchi 834001, Jharkhand, India; 2Department of Biotechnology, School of Biosciences & Technology, Vellore Institute of Technology (VIT), Tiruvalam Road, Vellore 632014, Tamil Nadu, India; 3Himalayan School of Biosciences, Swami Rama Himalayan University, Jolly Grant, Dehradun 248140, Uttarakhand, India; 4Institute of Advanced Materials, IAAM, Gammalkilsvägen 18, 59053 Ulrika, Sweden; 5Facultad de Medicina Humana, Universidad de San Martín de Porres, Chiclayo 15011, Peru; 6Department of Microbiology, All India Institute of Medical Sciences, Gorakhpur 273008, Uttar Pradesh, India; 7Department of Occupational Science & Occupational Therapy, Temerty Faculty of Medicine, University of Toronto, Toronto, ON M5G 1V7, Canada; 8Center for Transdisciplinary Research, Saveetha Dental College, Saveetha Institute of Medical and Technical Sciences, Saveetha University, Chennai 600077, Tamil Nadu, India; 9Department of Community Medicine, Faculty of Medicine, Datta Meghe Institute of Medical Sciences, Wardha 442107, Maharashtra, India; 10Joint School of NanoScience and Nano Engineering, University of North Carolina, Greensboro, NC 27402-6170, USA; 11Department of Community Medicine and School of Public Health, Postgraduate Institute of Medical Education and Research, Chandigarh 160012, Punjab, India; 12Tribhuvan University Teaching Hospital, Institute of Medicine, Kathmandu 44600, Nepal; 13Dr. D.Y Patil Medical College, Hospital and Research Centre, Dr. D.Y.Patil Vidyapeeth, Pune 411018, Maharashtra, India

**Keywords:** Ebola virus disease (EVD), Ebola vaccines, Ebola outbreak, challenges

## Abstract

The global outgoing outbreaks of Ebola virus disease (EVD) in different regions of Sudan, Uganda, and Western Africa have brought into focus the inadequacies and restrictions of pre-designed vaccines for use in the battle against EVD, which has affirmed the urgent need for the development of a systematic protocol to produce Ebola vaccines prior to an outbreak. There are several vaccines available being developed by preclinical trials and human-based clinical trials. The group of vaccines includes virus-like particle-based vaccines, DNA-based vaccines, whole virus recombinant vaccines, incompetent replication originated vaccines, and competent replication vaccines. The limitations and challenges faced in the development of Ebola vaccines are the selection of immunogenic, rapid-responsive, cross-protective immunity-based vaccinations with assurances of prolonged protection. Another issue for the manufacturing and distribution of vaccines involves post authorization, licensing, and surveillance to ensure a vaccine’s efficacy towards combating the Ebola outbreak. The current review focuses on the development process, the current perspective on the development of an Ebola vaccine, and future challenges for combatting future emerging Ebola infectious disease.

## 1. Introduction

It has been more than 40 years since the first Ebola outbreak occurred in 1976, which affected southern Sudan and the northern region of the Democratic Republic of Congo and lasted for almost 4 months, extending from June to September, with a mortality rate of 53% and 88.1%, respectively [1]. The two viruses that were later identified and held responsible for the first outbreak were Sudan ebolavirus and Zaire ebolavirus [2]. It is noticeable that most of the cases throughout have been predominant in central Africa. The countries that are majorly impacted by the Ebola outbreak include Sudan, Uganda, Ivory Coast, and the Democratic Republic of Congo [3]. Recently, on 20 September 2022, the Sudan ebolavirus-originated Ebola virus disease (EVD) outbreak desolated Uganda; the Ministry of Health reported fifty-five fatal confirmed deaths to the World Health Organization through 21 November 2022. However a similar outbreak caused by Sudan ebolavirus was previously reported in 2012 as well. In Sudan, EVD was caused due to Zaire ebolavirus that was imported from the Democratic Republic of the Congo (DROC). Recently, the DROC reported Ebola outbreaks on 23 April 2022 and 4 July 2022, which was followed by a third outbreak that occurred in 2018. Table 1 summarizes the epidemiological outbreaks of EVD since 1976. Fortunately, the preparedness against these outbreaks regulated by vigorously active national authorities has dynamically restricted the transmission of the Ebola virus. However, these past outbreaks have provided present and future epidemiologists and researchers with critical situations and lessons which can infor m preparations and advisory measures against various Ebola outbreaks on a global platform. The CDC and WHO are partnering with a large number of communities from various countries at global platforms to support the rigorous reporting and examination of Ebola outbreaks and to reinforce the rapid and swift response against Ebola-mediated infectious diseases that concern a threat to public health [4].The pathogenesis of EVD is summarized in Figure 1.

EVD outbreaks cause mortality ranging from 50% to 90%; however, no specified antiviral drugs are required for the treatment, which indicates the control of an outbreak by earlier identification through symptoms with early medical care. The current Ebola outbreak indicates an urgent requirement for a vaccine against EVD.

The rapid and detailed scrutinization of vaccine development strategies from a technical perspective with respect to preceding vaccines used in Ebola may serve as preparedness against this re-emerging infectious disease. The issues of clinical study approval, the safety of vaccines, virus variants surveillance for vaccine development, rapid synthesis, and distribution of vaccines is a challenging, difficult, and time-consuming process, as it requires authorization and the attainment of licenses from big pharmaceutical companies. Other social factors, such as urbanization, mobility of individuals from one country to another, environmental factors, ecological equilibrium, and immunity factors, have an influence on viral infection reemergence and their transmission through zoonosis [5,6]. The approach of viral detection and regular monitoring of these outbreaks among communities, along with the management of Ebola vaccine development, will act as a combinatorial approach to combat such remerging outbreaks on a global level.

## 2. An Overview of Current Vaccines

### 2.1. Virus-Like Particles Vaccines

Zaire Ebolavirus (ZEBOV) matrix protein VP40 and glycoprotein (GP) make up virus-like particles (VLPs) with the occasional presence of nucleoprotein (NP). The creation of ZEBOV-like particles and cell budding is caused by the expression of VP40 in cells. The inclusion of these proteins into the VLPs is triggered by the expression of GP and/or NP. In mouse and guinea pig efficacy tests, VLPs made of VP40 and GP provided 100 percent protection from deadly ZEBOV infection [6]. Three vaccinations of non-human primates (NHPs) with VLPs containing glycoprotein, nucleoprotein, VP40, and an adjuvant (RIBI) resulted in immunological responses in the animals which were protective against deadly ZEBOV [7,8,9].

Immunogenic VLPs-based immunization triggers innate, humoral immune (HI), and cellular immune (CI) responses. Researchers turned to a baculovirus-based expression system employing insect cells to be able to expand VLP production, which was only performed in a limited amount in 293T cells. The immunogenic effect of the VLPs produced in these insect-based cell lines has been demonstrated in mice when used with the adjuvant QS-21; however, it is yet unknown if they are effective in protecting NHPs against the deadly ZEBOV challenge. ZEBOV and Marburg virus (MARV) were tested in guinea pig animal models utilizing chimeric VLPs and vice versa for the cross-protection as well as a combination of ZEBOV- and MARV-like particles on this platform. The findings confirmed the necessity of GP for protection and the efficacy of blended VLPs over chimeric VLPs [7,8,9].

### 2.2. DNA Vaccines

The first effective DNA vaccination method against ZEBOV, containing four doses of either ZEBOV-GP or ZEBOV-NP, was disclosed in 1998, demonstrating 100% protection from EBV in the vaccinated mice models. DNA vaccines have the benefit of being quickly adaptable as pathogens change, and the fact that plasmids are non-infectious and simple to create in vast quantities makes them particularly advantageous regarding new and re-emerging infections. Additionally, this strategy is reusable because pre-existing immunity is irrelevant. DNA vaccines elicit CMI and HMI, necessitating the administration of multiple doses to produce the desired immunity [10].

Later, it was shown that three dosages of plasmid DNA for strain 13 guinea pigs had partial protective efficacy. Notably, 50% of the mice that survived acquired viremia. Data on DNA vaccination alone in NHPs are lacking, but DNA paired with immunization using recombinant Adenovirus 5 (rAd5)-based vectors was successful. The phase I clinical trial demonstrated successful immunogenicity developed from three doses of a DNA vaccine encoding ZEBOV (GP or NP) and Sudan ebolavirus (SEBOV-GP) in human populations. The 20 individuals experienced zero side effects post vaccination, with the production of specific antibodies and CD4+ T cell responses against each vaccine component. Eight individuals confirmed the presence of vaccine-specific produced CD8+ T cells in their samples [11].

The DNA vaccination method for filoviruses has recently been enhanced in which plasmid vectors express codon-optimized antigens, enabling the injection of higher DNA concentration-based doses. The experimental animals were further exposed to a lethal dose of MA-ZEBOV after receiving two or three multiple combinatorial doses of ZEBOV-GP, SEBOV-GP, and MARV-GP antigens encoding plasmids or an individual antigen encoding plasmid. The ZEBOV-GP DNA vaccination provided protection to every animal that received it. By using ELISA, antigen-specific IgG responses were found. Although intramuscular electroporation was the most effective method, the magnitude of the humoral reaction was not significantly influenced by the vaccination doses of 1, 5, or 20 mg DNA. More recently, guinea pigs were reported to be rescued from the deadly GPAZEBOV challenge through prime/boost strategy with combinatorial approach by means of plasmids encoding ZEBOV-GP, SEBOV-GP, and MARV-GP [12,13]. Codon optimization has been proven to boost mRNA half-life, amplify translation efficacy, and stimulate gene expression. IM-EP immunization with monovalent filovirus DNA vaccines with codon optimization shielded numerous non-human primates against homologous EBOV challenges [14]. Following IM-EP, a synthetic plasmid vaccination encoding a GP consensus sequence from various West African EBOV-Makona strains demonstrated comparable effectiveness in non-human primate models [15]. Conversely, quadrivalent filovirus vaccination developed by Grant-Klein and co-workers could only impart protection against MARV, with just one-fifth of models sustaining the EBOV challenge [15]. The justification(s) for this lack of protection has yet to be determined. To assess the safety efficacy, and immunogenicity of DNA vaccines against EBVD, some phase I clinical trials have been performed that shown well-tolerance and induction of both humoral and cell-mediated immune response in human candidates [16,17,18,19]. However, further research and more clinical studies are still needed to implement the EBOV DNA vaccine in clinical applications. 

### 2.3. Recombinant Whole Virus Vaccine

The host immune system may respond more broadly and vigorously to whole virus vaccinations than to targeted vaccines that only deliver a single viral protein because they present the host immune system with many viral proteins as well as the viral genetic material. However, early attempts to create an EBOV vaccine that had been gamma-irradiated and rendered inactive failed to effectively protect nonhuman primates from a deadly dosage of EBOV challenge. An incompetent defective EBOVΔVP30 without crucial viral transcription activator from wild type VP30 (Mayinga strain of EBOV), was created by Marzi and colleagues. High titers of EBOVΔVP30 replication, genetic stability, and lack of pathogenicity in rodents all occur in cell lines that persistently express the VP30 protein. When challenged with lethal doses of EBOV that were mouse- or guinea pig-adapted, mice and guinea pigs that had received two doses of EBOVΔVP30 were completely protected [20,21].

Marzi and his colleagues reported the genomic stability of the EBOVΔVP30 was checked and no evidence of recombination or mutation of the VP30 deletion site was observed. In order to address this safety risk, rZEBOVDVP30 was serially passaged through Vero cells that expressed VP30. Groups of cynomolgus macaques were intramuscularly administered a single dose of 10^7^ focus forming units (FFU) of EBOVΔVP30, or two doses of 10^7^ FFU of EBOVΔVP30 four weeks apart in order to test the efficacy of EBOVΔVP30, a whole virus vaccine, in NHPs. However, the safety associated with the vaccine is of huge concern, leading to the need for viral inactivation using hydrogen peroxide. In order to restore vaccination safety, the EBOVΔVP30 viruses were treated with 3% H_2_O_2_, and subsequent viral plaque assay in VP30-expressing cells exhibited complete inactivation. However, animals administered with untreated or H_2_O_2_-treated EBOVΔVP30 showed no signs of illness [21].

### 2.4. Replication Incompetent Vaccines

The protection of non-human primates against lethal challenge with individual homologous viruses was reported to be regulated by numerous filovirus antigens comprising complex or blended vaccination [22]. Cross-protection against BEBOV was accomplished using ZEBOV and SEBOV GPs through rAd5-expression along with a complex or blended vaccination strategy. These findings show that it is theoretically conceivable to develop cross-protective immunity against many filovirus species.

In a phase I clinical trial, the rAd5 vaccine-expressing ZEBOV-GP was evaluated and found to be immunogenic and safe. The dosage-dependent T cell responses against antigen-specific molecules showed insufficiently elicited immune responses, thus, the lower level of protection they provided was insufficient [23]. In a previous investigation, CD8+ T cell responses were found to be linked with NHP protection from the ZEBOV-GP rAd5 vaccine. However, more recently, Wong et al. examined ZEBOV-GP-specific T cell immunity and antibody responses in rats and NHPs inoculated with rAd5 vaccines and discovered that ZEBOV-GP-specific IgG, not T cell immunity, was the essential component for protection [24].

### 2.5. Replication Competent Vaccines

Based on the vaccinia virus (VV), one of the first platforms used to create an EBOV vaccine, homologous DNA recombination was used to create VV-based vaccines, and in the case of EBOV, several ZEBOV genes were selected as single antigens: GP, solube GP (sGP), NP, the polymerase co-factor VP35, and VP40. VV-based vaccines administered in strain 13 guinea pigs did not cause viremia; however, the same vaccine administration in NHPs resulted in viremia, and the subjects were euthanized [25].

Recombinant murine cytomegalovirus (CMV) was genetically modified to express a CTL epitope found on ZEBOV-NP (amino acids 43–54) by fusing it to the ie2 gene in a “proof-of-concept” experiment. After a single mouse immunisation, CTL responses to the ZEBOVNP epitope were easily identified. After receiving two doses of recombinant murine CMV/ZEBOV-NP vaccination, C57BL/6 mice were challenged with a lethal dosage of MA-ZEBOV. The immunised mice survived the test but lacked MA-ZEBOV replication protection. These findings suggest a protective role for CTL responses against deadly ZEBOV infection, but they must be confirmed in further animal models utilising species-specific CMVs (e.g., macaques) [26].

Negative-stranded RNA viral vectors have been used in a number of strategies in addition to DNA virus-based vaccinations. Based on the human parainfluenza virus 3, recombinant HPIV3 (rHPIV3) has been studied as a dual vaccine strategy against HPIV3 and measles infections in babies because it is a frequent respiratory disease. ZEBOV-GP and/or ZEBOV-NP were added to rHPIV3 to enable ZEBOV vaccination. Each vector, rHPIV3/ZEBOV-GP or rHPIV3/ZEBOV-GP/NP, was administered to guinea pigs intranasally once, and this was enough to shield all the animals from deadly disease. Two vaccination doses were necessary for rhesus macaques to receive 100% protection when administered through the respiratory tract [27,28].

The reverse genetics approach created for the vesicular stomatitis virus (VSV), a prototypical member of the Rhabdoviridae family, is the foundation of a very promising vaccine platform for EBOV. The VSV glycoprotein (G), the viral determinant for neurotropism and pathogenicity, is absent from the current EBOV vaccine vector, which resembles an attenuated form of VSV, the Indiana serotype. In the general population, pre-existing immunity is scarce and, when it does exist, it is usually directed against VSV-G, which is absent in this vector. Instead of VSV-G, the rVSV/ZEBOV vector encodes ZEBOV-GP, a highly immunogenic vaccine virus that is attenuated but still easily propagable and causes only temporary vector viremia in inoculated patients [29].

The current vaccine scenario is summarized in Figure 2 and Table 2.

## 3. Human Clinical Trials of Ebola Vaccines

The reappearance of the Ebola virus epidemic in Guinea intensified the demand for an efficient and secure immunization to prevent further outbreaks. [34] The latest EVD (Ebola virus disease) vaccination strategies target EBOV, the leading variant of the Ebola virus, for which numerous phase 1 to phase 4 human clinical trials were conducted in recent years [39]. Phase 1 trials involve a handful of healthy individuals to analyze the detrimental impacts of the vaccine incorporation. In the phase 2 trials, a larger population is used to check the safety and immunogenicity of the vaccines formulated. Phase 3 studies evaluate the vaccine’s effectiveness during an epidemic, and phase 4 trial studies are used to monitor a vaccine’s safety after it has been released onto the market [40]. There are more than 70 clinical trials for vaccination against EBOV virus, out of which, only 2–3 received FDA approval by 2020. The vaccines worth mentioning that have successfully prevented further spread of the disease are the recombinant VSV-based vaccine (VSV-EBOV), Ad26-ZEBOV/MVA-BN-Filo vaccine, and GamEvac-Combi vaccine. The currently available vaccine candidates against EBOV virus include replicative vectored vaccines, non-replicative vectored vaccines, polypeptide vaccines, protein nanoparticle vaccines and DNA vaccines [40].

The ability of virus-vectored vaccines to direct antigens, particularly to target cells, and to establish strong, long-lasting immunity makes them the most suitable type out of all. The GamEvac-Combi vaccine is a replicative vectored vaccine having a combination of VSV-EBOV and Ad5-EBOV vaccines merged into a single immunization approach, where Ad5-EBOV serves as a stimulant to the core VSV-EBOV vaccine. Initial phase 1 and 2 human trials were conducted on healthy individuals in Russia as an accessible, heterologous prime-boost trial with dosage progression study. The two controls administered were either VSV-EBOV or Ad5-EBOV vaccine only, whereas the heterologous groups were administered either half or full dosage of both vaccines. The half-dosage homologous group had comparatively more severe reactions like fever, even though there was not much difference observed in the nature and adversity of the reaction, thus demonstrating the safety of the vaccine [41]. Further, the Russian Federation’s Ministry of Health licensed the vaccine for use in humans. Phase 4 trials have been conducted in Guinea and Russia on a population of 2000 individuals to assess the efficiency, safety, and ability of the virus to initiate an immunogenic response [40]. The European Commission approved the marketing authorization for the Janssen Vaccines & Prevention heterologous primary and booster Ebola vaccine regimen in May 2020 to combat Ebola virus infection in both adults and children of atleast 1 year age. This regimen includes the type 26 adenovirus vector-based vaccine that expresses Zaire Ebola virus glycoprotein (Ad26.ZEBOV) and the modified vaccinia Ankara vector-based vaccine, which encodes glycoproteins from multiple viruses like Zaire ebolavirus, Sudan virus, and Marburg virus, and also the nucleoprotein from Tai Forest virus (MVA-BN-Filo). This non-replicative vectored vaccine demonstrated high efficacy and safety profile after the trials were conducted [34,39]. The recombinant VSV-based vaccine (VSV-EBOV), which is a replication-competent vaccine, had multiple phase 1 clinical trials all over the world during the 2013–2016 epidemic. In two initial trials for the VSV-EBOV, one group received a second dosage after a 28-day gap, and the trials were non-randomized, double-blind, and dosage-escalation trials. Two out of the sixty people who received the vaccine experienced a transient grade 3 fever with a temperature of 39–40 °C, the adversity of which decreased in a dosage-based manner. A small number of individuals experienced grade 3 severe reactions between 12 and 24 h after immunization, which included myalgia, tiredness, and headaches. Additionally, a few vaccine recipients experienced some unexpected side effects (oral ulcers, infectious colitis, and cervical lymphadenopathy) that appeared but went away without any problems. Of the 60 volunteers who received the immunization, 19 experienced severe arthralgia within 7 days, although arthritis did not progress. Later phase 2 trials took place in Sierra Leone and Liberia. A phase 2 clinical trial found that 83.7% of vaccinations produced EBOV GP-specific antibody responses, in comparison to 2.8% in the control group, as assessed by ELISA, which lasted for 12 months. Guinea, the U.S., Spain, Canada, the DRC, and Sierra Leone have all conducted phase 3 studies. In the massive 2015 Guinea ring immunization cluster-randomized study, 5837 participants were randomly assigned to receive the 2 × 107 plaque-forming units (pfu) VSV-EBOV vaccine either instantaneously or thereafter. The trial revealed 100% efficiency in the instantaneously administered vaccine group. Three different vaccination lots were examined in a batch consistency phase 3 research in the United States, Spain, and Canada; no safety issues were identified. The VSV-EBOV was authorized for emergency use based on the positive clinical trial findings. A lot of research still needs to be conducted in order to develop better vaccines against the Ebola virus, given that few of these vaccines were able to complete phase 3 or phase 4 clinical trials due to multiple difficulties and challenges encountered during the process [42].

## 4. Animal Models for EVD Vaccine Development

Historically, vaccine prospects for EBOV, as well as other filoviruses, have been tested in rodent models such as mice, hamsters, or guinea pigs [43] (Table 3). However, none of the experimental filovirus variants elicited infection in these rodent models. As a result, sequential acclimation is usually necessary to achieve equal lethality in rodents. Adaptation is connected with genomic alterations, which arise frequently in genes encoding interferon inhibitors, including VP24 for EBV and VP40 for Marburg viruses. Rodent species that employ modified filoviruses may not accurately replicate human clinical signs and development, particularly mice models. However, despite the fact that overall efficacy in non-human primates and humans may not be accurately predicted by them, rodents are still considered popular testing models [44,45]. Due to clinical manifestations identical to those found in humans, non-human primate models are still regarded as the benchmark for filoviruses [45]. Cynomolgus macaques are chosen for preventive vaccination research, although rhesus macaques are commonly employed for pharmaceutical investigations, owing to their longer duration before death, which provides for a longer timeframe for interventions [42]. The majority of EBOV vaccine candidates now in development have undergone rodent testing, non-human primate validation assessment, and clinical studies.

## 5. Challenges for Vaccine Development

### 5.1. Selection of Immunogen

For the development of viral vaccines, either the entire pathogens (live or attenuated) or their membrane protein subunits, polysaccharides, are used together with the adjuvants to enhance and elicit the immune reaction. The fundamental method for the choice of immunogens for the Ebola vaccine is humoral pathways (set off neutralizing antibodies) and cellular-mediated pathways (function of T cells) [46]. The discovery of a selectable antigen as an immunogen for Ebola vaccine development is a major challenge for disorder control. In general, for the deactivation of immunoglobins or and anti-monoclonal antibodies (ZMapp), the transmembrane glycoproteins are used to prevent and facilitate recovery of contamination in non-human primates. The Ebola virus’s surge of highly glycosylated glycoproteins (EBOV-GP-1,2) are the target for the generation of the humoral immune response in the host body. In the pre-clinical trial, Sheep were immunised with genetically engineered EBOV-GP ectodomain (EBOV-GP1,2ecto) expressed in mammalian cells, which resulted in a potent immunological response and the production of high titres of high avidity polyclonal antibodies [47]. Besides that, the type 1 Viral GP act synergistically to bind as trimeric spikes at the external side of the virion envelope for the cell receptor binding, thereby consequently combatting the invasion of the virus in the host cell body. Therefore, GPs are selected as most suitable target for the host immune response and are applicable as a conjugant for the design of a viral vaccine. The recently approved vaccine Ervebo, a recombinant vector-based Ebola virus vaccine, was designed on the ebolavirus GP by targeting the three immunogenic peptides sequences (P1—FKRTSFFLWVIILFQRTFSIPL, P2—LANETTQALQLF, and P3—RATTELRTFSILNRKAIDF) for the human leukocyte antigen (HLA), respectively. These GPs simulate the proliferation of peripheral blood mononuclear cells and the synthesis of interferon-gamma (IFN-γ) molecules [48].

### 5.2. Rapid-Responding Vaccination

Several vaccines have been examined in phase 1 studies and clinical trials. The two most advanced first-generation Ebola vaccine candidates are alive-replicating vesicular stomatitis virus (rVSV) and the attenuated chimpanzee adenovirus 3 (ChAd3) [21]. For an effective outcome and rapid immunization, the structural and functional immunogens simply need to be studied by collecting the infected and immunizing potential contacts with an experimental Ebola vaccine. Consequently, the U.S. Food and Drug Administration (FDA) permitted clinical trial of efficient Ebola vaccine ERVEBO (Ebola Zaire Vaccine): V920 (rVSVG-ZEBOV-GP or rVSV-ZEBOV), which produced a rapid antibody response within 14 days after a single dose [49].

The basic approaches for the fast-acting vaccines should be permanent and prime-boosters for longer-term protection. The preventive vaccination strategy is based on the populations at risk, specifically for healthcare workers and frontline workers, based on the assessment of the durability of immunity; however, limited information or records were provided for 1 year post-vaccination. In addition, the rapidity of immune response induction is an important factor for estimating the relative effectiveness of a immunization from the perspective of ring vaccination. Hence, the clinical studies of the Ebola vaccines presented no cases of viral infection after vaccinations with respect to transmittance through randomized trails [22].

The association between host immunity and disease prevention as a factor to help cure viral infection is still unknown, and no correlation with protection has yet been found. However, medical data and case studies have verified the stability of viral-induced specific responses and fortification measures for the designed vaccine efficacy and immunogenicity in non-human primates. Additionally, the clinical trials focused on the elucidating the host immune response with respect to the pathogen, i.e., Ebola virus. Therefore, the correlation between protection efforts and host immune responses provide the insight of molecular level interaction for the design of a vaccine. For research studies of rapid-response vaccines, the effectiveness and longevity of the evoked immunity in hosts of different age groups, including pregnant ladies, are major issues to be considered before vaccination. The value-added scientific information about the humoral and cellular immune responses generated in the host after vaccination need to be examined to select protective measures. Responding to these interrogations requires improving global capacities to continue Ebola vaccine research and collaborative partnerships to optimize the chances of success.

### 5.3. Cross-Protective Immunity

Through the transfer of infected individual plasma for vaccination, cross-protective immunity plays a crucial part in the development of vaccines for the common viral illnesses. Despite serological cross-reactivity, the development of EBOV countermeasures has not been impeded by findings of inter-species cross-protective immunizations [22]. In the pre-clinical trials, there are two recombinant viral vaccines: SEBOV-GP and -VP40 showed improvement in the rate of cross-protection in ≥90% guinea pigs’ population with survivability against the ZEBOV [50]. Consequently, the double dose of viral vaccine containing single rVSV vector having both SEBOV-VP40 and SEBOV-GP the inter-species pathogenic protection found in the clinical trials against the African EBOV species: Zaire strain Ebola virus (ZEBOV) and SEBOV. Future research will examine whether combining immunogens from various EBOV species with elevated expression levels of allied antigenic molecule to improve cross-species protective immunity [29]. The ZEBOV GP was then used for the development of recombinant viral vector for the Ebola vaccine, resulting in rVSVG-ZEBOV-GP (rVSV-ZEBOV). In the phase I clinical trials for the ring vaccination, the rVSV-ZEBOV demonstrated to be immune evoker, non-toxic and harmless. However, the 24 h study showed in the Resus macaques infected with the Ebola virus Makona showed the post-vaccination 33–67% effectiveness [51].

Non-human primates developed cross-protective immunity after receiving a recombinant adenovirus serotype 5 boost and a DNA immunization. In pre- and post-exposure challenge experiments, a recombinant vesicular stomatitis vaccination vector protected non-human primates. Although there are now several potential vaccine candidates, it is still unclear what factors contribute to protection against EBOV infection. It should be possible to develop enhanced vaccinations that are efficacious as post-exposure treatments or vaccines that cross-protect against the four African EBOV by combining existing vaccine candidates with research into the correlation between protection and the use of genomics methods [51].

### 5.4. Long-Term Protection

The statistic showed five distinct species of the ebola virus and specific vaccinations only offer defense to guard against all the viruses, thus to make multimode viral vaccine for the viruses currently be a difficult approach. It is difficult, costly, and requires a lot of regulatory permission to create broad vaccines with several components [52]. The absence of cross protection offered by currently available vaccinations against heterologous species with similar genetic divergences implies that vaccines under development will not offer protection against newly developing Ebola viruses. Thus, in order to protect persons who are in danger of encountering the virus, a vaccine that can produce long-lasting protection is required, as virus epidemics are sporadic and impulsive. The prolonged effectiveness of vesicular stomatitis virus (VSV)-based vaccine (VSVG/EBOVGP) studies in rodents (mice and guinea pigs) provides a long-standing defense against Ebola viral infection, and proved supportive data for the non-human primates [53]. The prolonged two-phase study to monitor the effectiveness of Ebola vaccine for the respirational and sublingual (SL) adenovirus in non-human primates revealed durable protection from a single dose in monkeys with diverse Ebola GP-specific CD4+ T cells generation, and was an unstable approach [54].

### 5.5. Mechanism of Protection

In vivo animal model studies or trials provided diverse platforms for genomic, protein, or viral vaccine design since the development of the first Ebola vaccine, which concentrated on efforts to inactivate the virus. The pre- and post-exposure treatment plans include a variety of techniques, including DNA immunisation, RNA interference, polymeric delivery system, virus-like particles (VLPs), Venezuelan equine encephalitis virus replicons (VEEV RPs), serotypes, attenuation, and replication-competent viral platforms, e.g., human parainfluenzavirus 3 (HPIV3) and recombinant vesicular stomatitis virus (rVSV) [49].

Viral infection replication is particularly successful in inducing strong, long-lasting immune responses in a host. Contrarily, numerous genetically modified antigenic molecules as subunit DNA plasmids or proteins have commonly shown low immunogenicity, despite being thought to be generally safe (depending on the adjuvant). More than ten vaccine candidates are being developed; the two progressive vaccines are chimp adenovirus 3 (cAd3)-EBO Z(NIH-GSK) and recombinant vesicular stomatitis virus (rVSV)-ZEBOV, both of which are chimeric viral genome, along with the viral immunogenic glycoprotein (Canadian Dept. Public Health-Merck) used to induce immunogenicity. Likewise, in human primate models, for the initial identification of the cross-protection, the above-mentioned vaccines are quite immunogenic to threshold levels of antibody production on single dosage and cause a reduction in T-cell immunity.

There are a lot of data to support the importance of antibody responses. Among them, the ZMapp—humanised monoclonal antibody used as therapeutic molecule—showed promising outcomes in the preclinical models. However, the clinical trials of ZMapp in the West African population could not meet the standards and statistical threshold for effectiveness for the sample size due to the low frequency of EBOV with the experiment duration for long-term study [55].

Non-human primates (NHPs) were demonstrated to be protected after exposure to EBOV by pure sheep antibodies produced in contradiction of antigen, i.e., EBOTAb. Studies on passive transmission using human recovering sera have been utilized for further clinical studies. Additionally, EBOV typically results in acute illness, which is more frequently managed by antibodies, whereas chronic infections are typically better controlled by cytotoxic T-cells [47]. The identification of procedures that guarantee the security and efficiency of potential vaccines is advantageous for the regulatory licensing of vaccines. The global agency WHO recommends the design of orientation resources for EBOV pathology sero-testing and molecular analysis. Since these procedures commonly include bioassays, the ability to compare the results of the tests or analyses across periods or among labs depends on the accessibility of reference components to better diagnose and complement the data.

## 6. Current Vaccine Considerations

The platforms of vaccine development for the Ebola virus include nucleic acid vaccines, which are the mRNA and plasmid DNA vaccines, the viral vectored vaccines, which are the replicative and non-replicative viral vectored vaccines, and the protein-based vaccines, which are the recombinant subunit vaccine and the virus-like particle. The nucleic acid vaccines are translated to the target antigens either in the endosomes or in the nucleus. Replicative and non-replicative viral vector vaccines present the target antigen, the replicative ones keep the character of the virus, such as replication of the progeny virus particles and cell infections. The protein-based vaccines act by recognizing B-cells through the B-cell receptors and can be exposed to MHC by macrophages, dendritic cells, and and B-cells [56].

The current Ebola virus vaccines are INO-4201 (DNA vaccine), rVSVΔG-ZEBOV-GP, GamEvac-Combi, GamEvac-Lyo, rVSV N4CT1 EBOVGP1 (replicative vectored vaccine), Ad5-EBOV, Ad26.ZEBOV, MVA-BN-Filo, ChAd3, ChAd3-EBOZ, and MVA-VN-Filo, ChAd3, ChAd3-EBOZ, MVA-BN-Filo (non-replicative vectored vaccine), EpiVacEbola (polyepitope vaccine), and nanoparticle recombinant Ebola GP vaccine (protein nanoparticle vaccine) [40].

In 2019, one vaccine was approved by the FDA that had completed a phase 3 clinical trial, and two other drugs were approved by the FDA in 2020. The vaccines are ERVEBO^®^ (rVSV-ZEBOV-GP, v920), GamEvac-Combi (vaccine), Zabdeno and Mvabea (Ad26.ZEBOV, MVABN-Filo), and Ad5-EBOV. Ervebo (rVSV-ZEBOV) is a live and attenuated form of a vaccine that is made by genetic modification of the protein from the Zaire EBOV [30]. GamEvac-Combi is a VSV-Ad5 prime-boost EBOV vaccine [41]. The monovalent Ad26-ZEBOV vaccine provides acquired immunity to the Zaire EBOV, which is active and specific [57]. The recent vaccines that are in clinical trials are Ad26.ZEBOV, MVABN-Filo, rVSV#GZEBOV-GP from Zaire EBOV, which was developed in the countries of Guinea, Liberia, Mali, and Sierra Leone and is in phase 2 of the clinical trial, with the status of active and not recruiting (2017–2024) [58], GamEvac-Lyo, GamEvac-Lyo (component A), GamEvac-Lyo (component B) from Zaire EBOV, which is in the country of the Russian Federation and has completed phase 1 and phase 2 of the clinical trial (2017–2018) [41], VSV-GZEBOV, ChAd3-EBO Z from Zaire EBOV, which was developed in the country of Liberia and is in phase 2 of the clinical trial (2015–2020) [59], HPIV3/#HNF/EbovZ GP vaccine from Zaire EBOV, which was developed in the country of the USA and is in phase 1 of the clinical trial (2018–2020) and is in active and not recruiting stage, ChAd3-EBO-Z, MVA Multi-Filo Ebola vaccine from Mayinga EBOV, which was developed in the USA and has completed phase 1 of the clinical trial (2018–2020) [16], cAd3-EBO S vaccine from Zaire EBOV (Mayinga), which was developed in Uganda and has completed phase 1 of the clinical trial (2019–2020) [34], and cAd3-Marburg cAd3-EBO-S from Zaire EBOV (Mayinga), which was developed in the USA and is in recruiting phase 1 of the clinical trial (2021) [34].

The live attenuated vaccines have several drawbacks. On the contrary, multi-epitope vaccines are safer. Employing an immunoinformatics approach to design such a multi-epitope vaccine against EBOV is now considered a novel approach for designing safer drugs. The vaccines thus designed possess all the desirable qualities of solubility, high antigenicity, and non-allergenicity [56]. The immunoinformatics approach helps in the rapid and specific screening of all possible epitopes with high immunogenic potential. This helps with the identification of the immunogenic peptide fragments. These potent peptide vaccines can be verified in animal models and also be used for designing efficacious synthetic Ebola vaccines [48]. Multivalent vaccines have been developed, but specific evaluation of the immune response or efficacy against other Ebola virus species has not yet been done. Future vaccine development should focus on the aspect of protection against all relevant species of the virus, as intraspecies mutations may impact the effectiveness of the vaccines [34]. Plant-based vaccines against the Ebola virus are also unexplored, although some are under clinical evaluation. The areas which need special attention for the production of plant-based vaccines are stable nuclear expression, chloroplast expression, and viral-vector-mediated transient expression [60]. There are also many repurposed compounds that can act as inhibitors of Ebola virus. In depth study of the pathogenesis and role of these compounds are required to understand the host–pathogen interaction in the EBVD and to facilitate the development of vaccines accordingly.

## 7. Conclusions

Ebola vaccine development has shown and proven remarkable progress in preclinical and clinical phases. These vaccines emerged as multiple targeting potential candidates in their advanced stages. However, the obstructions and challenges related to the efficacy, potency, durability, and cost-effective methodologies in development of Ebola vaccines still need to be properly addressed. Contrary to the previous Ebola vaccines, the current vaccines require a relatability of immunological response of an individual, epidemiological data, and clinical trial results in the community with respect to the vaccine’s efficacy. On these bases, potential vaccines can be developed and applied to combat remerging infectious diseases that may cause future Ebola infections. This approach requires strong monitoring, observance, investigation, and preparedness among the researchers, epidemiologists, vaccine-developing pharmaceutical organisations, stake-holders, and funders on a global level.

## Figures and Tables

**Figure 1 vaccines-11-00268-f001:**
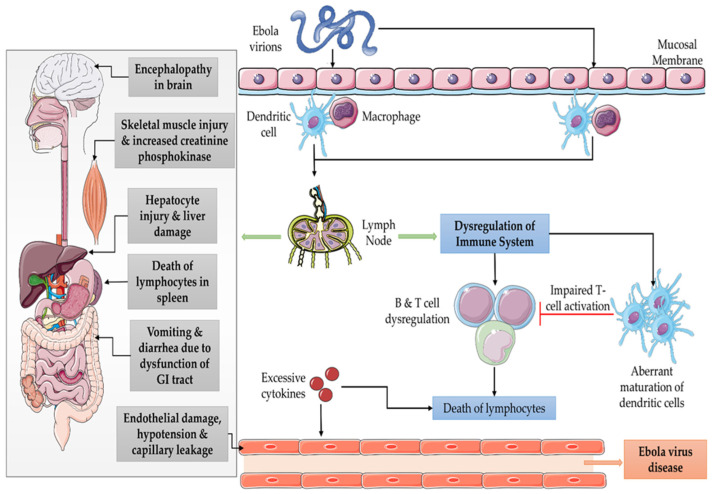
Diagrammatic representation of pathogenetic mechanism adopted by EVD.

**Figure 2 vaccines-11-00268-f002:**
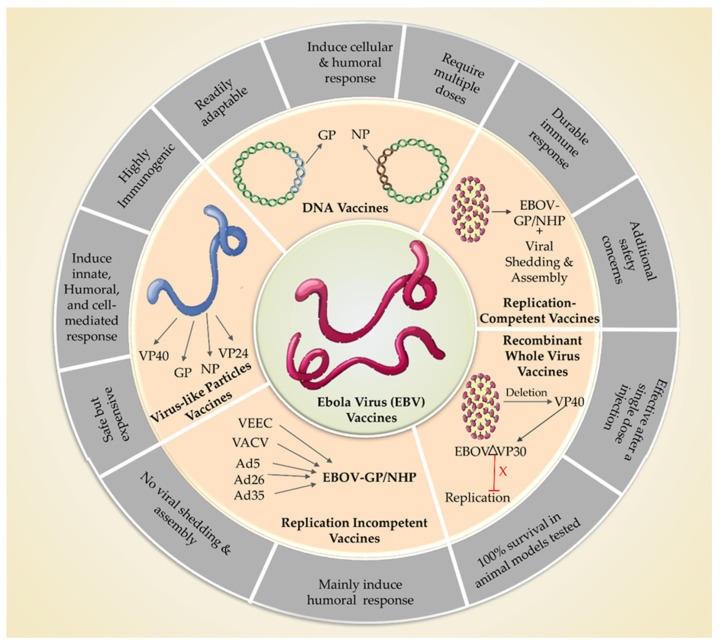
Current vaccination platforms and strategies available for EVD.

**Table 1 vaccines-11-00268-t001:** Epidemiological outbreaks of EVD since 1976 (https://www.who.int/news-room/fact-sheets/detail/ebola-virus-disease, accessed on 8 January 2023).

Year	Country	EVD	Cases	Deaths	Case Fatality
2021	Guinea	Zaire	Ongoing		
2021	Democratic Republic of the Congo	Zaire	Ongoing		
2020	Democratic Republic of the Congo	Zaire	130	55	42%
2018–2020	Democratic Republic of the Congo	Zaire	3481	2299	66%
2018	Democratic Republic of the Congo	Zaire	54	33	61%
2017	Democratic Republic of the Congo	Zaire	8	4	50%
2015	Italy	Zaire	1	0	0%
2014	Spain	Zaire	1	0	0%
2014	UK	Zaire	1	0	0%
2014	USA	Zaire	4	1	25%
2014	Senegal	Zaire	1	0	0%
2014	Mali	Zaire	8	6	75%
2014	Nigeria	Zaire	20	8	40%
2014–2016	Sierra Leone	Zaire	14,124	3956	28%
2014–2016	Liberia	Zaire	10,675	4809	45%
2014–2016	Guinea	Zaire	3811	2543	67%
2014	Democratic Republic of the Congo				
2012	Democratic Republic of the Congo	Bundibugyo	57	29	51%
2012	Uganda	Sudan	7	4	57%
2012	Uganda	Sudan	24	17	71%
2011	Uganda	Sudan	1	1	100%
2008	Democratic Republic of the Congo	Zaire	32	14	44%
2007	Uganda	Bundibugyo	149	37	25%
2007	Democratic Republic of the Congo	Zaire	264	187	71%
2005	Congo	Zaire	12	10	83%
2004	Sudan	Sudan	17	7	41%
2003 (Nov–Dec)	Congo	Zaire	35	29	83%
2003 (Jan–Apr)	Congo	Zaire	143	128	90%
2001–2002	Congo	Zaire	59	44	75%
2001–2002	Gabon	Zaire	65	53	82%
2000	Uganda	Sudan	425	224	53%
1996	South Africa (ex-Gabon)	Zaire	1	1	100%
1996 (Jul–Dec)	Gabon	Zaire	60	45	75%
1996 (Jan–Apr)	Gabon	Zaire	31	21	68%
1995	Democratic Republic of the Congo	Zaire	315	254	81%
1994	Côte d’Ivoire	Taï Forest	1	0	0%
1994	Gabon	Zaire	52	31	60%
1979	Sudan	Sudan	34	22	65%
1977	Democratic Republic of the Congo	Zaire	1	1	100%
1976	Sudan	Sudan	284	151	53%
1976	Democratic Republic of the Congo	Zaire	318	280	88%

**Table 2 vaccines-11-00268-t002:** Vaccines against Ebola that are currently approved and under development.

Vaccine Name	Manufacturer	Type/Category	Other Components	Present Status	Disadvantages and AE	Reference
rVSV-ZEBOV-GP; V920; rVSVAG-ZEBOV-GP (ERVEBO^®^)	Merck	Live, attenuated vaccine, Recombinant with vesicular stomatitis virus (rVSV). replication-competent	Monovalent, expresses EBOV Glycoprotein (GP) (Kikwit variant)	Approved by US FDA for 18 years and older	Only targets EBOV, which was responsible for the 2013–2016 outbreaks and more recent flareups Reports of arthritis as AE Synovial joints of vaccinated individuals’ reports finding of infectious virus causing secondary spread Stringent storage temperature	[30]
Ad26.ZEBOV MVA-BN-Filo boost (Zabdeno/Mvabea)	Johnson & Johnson (Janssen facility)/ Bavarian Nordic	Based on human adenovirus serotype 26 (Ad26)	Multivalent, EBOV GP, TAFV NP, SUDV GP, and MARV GP	Licensed by EMA (exceptional circumstances); Submission g to WHO	Non ideal candidate as per immunogenicity Mvabea vaccine shows lack of immunogenecity against *Bundibugyo* or *Bombali ebolaviruses*	[31]
Ad5-EBOV	BIT CanSino (China	Recombinant vaccine based on human adenovirus serotype 5 vector (Ad5)	Monovalent, expresses EBOV GP (Makona variant)	Approved by China Food and Drug Administration (CFDA) (2017) based on animal rule	EBOV specific; prior immunity to Ad5 decreases the effectiveness Suitable for 18 to 60 years of age; Lack of clinical data; antibodies decline 85% at day 168	[32]
ChAd3-EBOZ (+/−) Mvabea (cAd3-ZEBOV/ChAd3-EBO-Z)	GlaxoSmithKline, NIAID, Okairos	Recombined with attenuated version of a chimpanzee adenovirus (cAd3) unable to replicate in human,7	Monovalent, expresses EBOV GP (Mayinga variant)/Mvabea expresses EBOV, SUDV, MARV GP’s and TAFV NP	Not yet licensed by the US FDA or EMA Phase II trials completed in Europe, the US, and Africa	Enhanced vaccines doses for immunogenicity; antibody titre declines by roughly 50% at 180 days after vaccination; booster needed; storage condition issues	[33]
GamEvac-Combi and GamEvacLyo	Gamaleya Research Institute of Epidemiology & Microbiology (Russia)		Heterologous primary booster (rVSV and Ad5) expressing EBOV GP	Licensed under Ministry of Health of Russian Federation	prime + booster at 21 days both required; Age limitation from 18 to 55 years age; Un- published safety and efficacy. Preexisting neutralizing Ad5 GP responses in half-dose only	[34]
HPIV3/ΔHNF/EbovZ GP vaccine	NIAID	Live-Attenuated Human Parainfluenza Virus Type 3 Vectored Vaccine	Expressing Ebolavirus Zaire Glycoprotein as the Sole Envelope Glycoprotein			[34]
Rabies Vector-based GP vaccine	Thomas Jefferson University & NIAID	Recombinant vaccine based on rabies virus (RABV) as vector	Bivalent, Chemically inactivated RABV expressing EBOV GP	Non-human primate (NHP) challenge completed	In early developmental stage	[35]
EBOVΔVP30	University of Wisconsin	Hydrogen peroxide inactivated whole virus	based on a replication-defective EBOV (EBOVΔVP30)	NHP challenge complete	In early developmental stage	[36]
Vesiculovax	Auro Vaccines	Attenuated recombinant rVSV vector based	Expression of GP (Mayinga strain of Zaire ebolavirus)	Phase I	In early developmental stage	[37]
EBOV DNA Vaccine	NIAID	DNA vaccine	Encodes the envelope GP (Zaire &Sudan species) and the nucleoprotein	Phase I	In early developmental stage	[11]
EBOV DNA Vaccine	NIAID	DNA vaccine	Ebola virus (Zaire and Sudan) glycoproteins and (MAR) encoding Marburg virus glycoprotein	Phase 1b	In early developmental stage	[17]
Ebola Virus Glycoprotein Nanoparticle Vaccine	US Army Medical Research Institute of Infectious Diseases, Novavax	Recombinant nanoparticle vaccine	EBOV GP nanoconjugated	Phase 1	In early developmental stage	[38]

**Table 3 vaccines-11-00268-t003:** Animal models for development of EBOV Vaccines [+++: High; ++: Moderate; +: Low].

Ebola Virus	Animal Models	Route of Inoculation	Display of Human Symptoms	Cost of Handling	Ease of Handling	Applications
Wild Type EBOV	Non-human primate	Multiple routes of inoculation	+++	+++	+++	Confirmatory model
	Ferret	Intranasal	++	++	++	Not well established
	Humanized mouse	Intraperitoneal	++	+++	+	Not well established
Guinea Pig adapted EBOV	Guinea Pig	Intraperitoneal, Subcutaneous, Aerosol	+	+	+	Screening model
Mouse adapted EBOV	Laboratory mouse	Intraperitoneal, Intranasal, Aerosol	+	+	+	Screening model
	Hamster	Intraperitoneal	++	+	+	Screening model

## Data Availability

Not applicable.

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
