# Peer review of "Ebola Virus Disease Vaccines: Development, Current Perspectives & Challenges"

_vaccines, 2023, doi:10.3390/vaccines11020268_

Round 1

Reviewer 1 Report

I think the manuscript should be slightly revised before it is published

1.       Suggestions on current Ebola vaccine design strategies can be provided to enrich the content of the manuscript

2.     The epidemic history of Ebola can be described in more detail

Author Response

I think the manuscript should be slightly revised before it is published

Comment 1: Suggestions on current Ebola vaccine design strategies can be provided to enrich the content of the manuscript

Response:  Thank you for your suggestion. A section dedicated to current vaccine considerations is added (Section 5).

  1. The epidemic history of Ebola can be described in more detail

Response: Epidemic history of Ebola is described in more detail in the Introduction section. Furthermore, a table summarizing Epidemiological outbreaks of EVD since 1976 is added (Table 1).

Reviewer 2 Report

This review promises to provide up to date informations on Ebola virus disease vaccines: "Development, current perspectives and challenges". The reviewer assumes that vaccines are targeted to humans, even though animal models are required to develop them.

However, the review mainly focuses on old ideas and hopes. As an example is the review on DNA vaccines. There is no evidence that a protective immune response can be induced in humans using this technique. The hopes of DNA vaccines end around 2008 – 2013. It is expected that the authors are honest and resume after each chapter of a given vaccine technique – like DNA vaccine - if the vaccine is really inducing a protective immune response against Ebola virus in humans or how far the development in this aspect is.

For the (few) vaccines that showed protective immune responses in humans, it is expected that the authors respect the present safety aspect of a vaccine. Next, it may be fair to indicate against the Ebola viral strains, the vaccine was protective and finally what the missing viral strains are that need protection. 

In general, the literature cited is not up to date. This particularly true for some vaccine efforts.

This reviewer has the feeling that certain biases and favored vaccines are pertinent in this review. As a reader I expect to find up to date information on the state of the art on development – and their failure in human use if available as well as safety, protective effect and the phase (I- III) of vaccine trials in humans. This would meet the criteria the authors state themselves: “Development, current perspectives and challenges”

Author Response

Reviewer 2

This review promises to provide up to date information on Ebola virus disease vaccines: "Development, current perspectives and challenges". The reviewer assumes that vaccines are targeted to humans, even though animal models are required to develop them.

However, the review mainly focuses on old ideas and hopes. As an example is the review on DNA vaccines. There is no evidence that a protective immune response can be induced in humans using this technique. The hopes of DNA vaccines end around 2008 – 2013. It is expected that the authors are honest and resume after each chapter of a given vaccine technique – like DNA vaccine - if the vaccine is really inducing a protective immune response against Ebola virus in humans or how far the development in this aspect is.

For the (few) vaccines that showed protective immune responses in humans, it is expected that the authors respect the present safety aspect of a vaccine. Next, it may be fair to indicate against the Ebola viral strains, the vaccine was protective and finally what the missing viral strains are that need protection. 

In general, the literature cited is not up to date. This is particularly true for some vaccine efforts.

This reviewer has the feeling that certain biases and favored vaccines are pertinent in this review. As a reader I expect to find up to date information on the state of the art on development – and their failure in human use if available as well as safety, protective effect and the phase (I- III) of vaccine trials in humans. This would meet the criteria the authors state themselves: “Development, current perspectives and challenges”.

Response: Thank you for your suggestions. As per the suggestion, animal models for development of Ebola vaccines are described in Section 4.

The manuscript has summarized various types of vaccine candidates with their safety, immunogenicity, and clinical developments. Some clinical trials and drawbacks of DNA vaccines are described in Section 2.2. A section dedicated to current vaccine considerations is added as Section 5. The manuscript also presents data for clinical trials of different vaccines, and the vaccines that have been approved for clinical uses in particularly in section 2, 4, and 5. Furthermore, more recent literature are cited for up-to-date information.

Round 2

Reviewer 2 Report

This is a significantly improved and up to date version of the manuscript.